# A Narrative Review on Various Oil Extraction Methods, Encapsulation Processes, Fatty Acid Profiles, Oxidative Stability, and Medicinal Properties of Black Seed (*Nigella sativa*)

**DOI:** 10.3390/foods11182826

**Published:** 2022-09-13

**Authors:** Muhammad Abdul Rahim, Aurbab Shoukat, Waseem Khalid, Afaf Ejaz, Nizwa Itrat, Iqra Majeed, Hyrije Koraqi, Muhammad Imran, Mahr Un Nisa, Anum Nazir, Wafa S. Alansari, Areej A. Eskandrani, Ghalia Shamlan, Ammar AL-Farga

**Affiliations:** 1Department of Food Science, Faculty of Life Sciences, Government College University, Faisalabad 38000, Pakistan; 2National Institute of Food Science & Technology, University of Agriculture, Faisalabad 38000, Pakistan; 3Department of Nutritional Sciences, Faculty of Medical Sciences, Government College University, Faisalabad 38000, Pakistan; 4Faculty of Food Science and Biotechnology, UBT-Higher Education Institution, Rexhep Krasniqi No. 56, 10000 Pristina, Kosovo; 5Biochemistry Department, Faculty of Science, University of Jeddah, Jeddah 21577, Saudi Arabia; 6Chemistry Department, Faculty of Science, Taibah University, Medina 30002, Saudi Arabia; 7Department of Food Science and Nutrition, College of Food and Agriculture Sciences, King Saud University, Riyadh 11362, Saudi Arabia

**Keywords:** black seed (*Nigella sativa*), extraction methods, fatty acid composition, oxidative stability, encapsulation, food applications, health benefits

## Abstract

The current review investigates the effects of black seed (*Nigella sativa*) on human health, which is also used to encapsulate and oxidative stable in different food products. In recent decades, many extraction methods, such as cold pressing, supercritical fluid extraction, Soxhlet extraction, hydro distillation (HD) method, microwave-assisted extraction (MAE), ultrasound-assisted extraction, steam distillation, and accelerated solvent extraction (ASE) have been used to extract the oils from black seeds under optimal conditions. Black seed oil contains essential fatty acids, in which the major fatty acids are linoleic, oleic, and palmitic acids. The oxidative stability of black seed oil is very low, due to various environmental conditions or factors (temperature and light) affecting the stability. The oxidative stability of black seed oil has been increased by using encapsulation methods, including nanoprecipitation, ultra-sonication, spray-drying, nanoprecipitation, electrohydrodynamic, atomization, freeze-drying, a electrospray technique, and coaxial electrospraying. Black seed, oil, microcapsules, and their components have been used in various food processing, pharmaceutical, nutraceutical, and cosmetics industries as functional ingredients for multiple purposes. Black seed and oil contain thymoquinone as a major component, which has anti-oxidant, -diabetic, -inflammatory, -cancer, -viral, and -microbial properties, due to its phenolic compounds. Many clinical and experimental studies have indicated that the black seed and their by-products can be used to reduce the risk of cardiovascular diseases, chronic cancer, diabetes, oxidative stress, polycystic ovary syndrome, metabolic disorders, hypertension, asthma, and skin disorders. In this review, we are focusing on black seed oil composition and increasing the stability using different encapsulation methods. It is used in various food products to increase the human nutrition and health properties.

## 1. Introduction

Black seed (*Nigella sativa*) is an annual flowering plant in the Ranunculaceae family and Plantae kingdom. Black seeds are mostly found in western Asia, the Mediterranean North Sea area, and western and southern Europe. The black seed is also described in the Bible as the “healing black seed”, Hippocrates and Discroides termed it as Melanthion, and Pliny coined it Gith [1]. According to the world’s agricultural production, the production of oil seed is 40.29 million metric tons in Pakistan; in all the world, it is 607.3 million metric tons [2]. According to the Unani Tibb medical system, *Nigella sativa.* has proved very helpful in curing many health disorders. *Nigella sativa* has been used since ancient times in various civilizations of the world, and it is recommended as a “miracle cure” because it has the potential to cure several diseases and regulate the process of natural healing in the human body [3]. According to Indian medicinal culture, seeds can be consumed as a bitter, anthelmintic, astringent, jaundice, stimulant, intermittent fever, diuretic, paralysis, emmenagogue, piles, skin diseases, and dyspepsia [4,5]. They can be utilized in the form of an anti-cancerous, -diabetic, -bacterial, hepato-toxic, -parasitic, and -fungal, as well as a therapeutic agent. Black seeds in herbal medicines are consumed directly as an active ingredient or in the form of herbal tea. The black seed extract has the tendency to show anti-oxidant and -inflammatory properties. It has been used by patients to suppress coughs, disintegrate renal calculi, impede the carcinogenic process, treat abdominal pain, diarrhea, flatulence and polio, exert choleretic and uricosuric activities [4,6]. According to former literature, *Nigella sativa* seeds show various properties against different kinds of cancer, such as blood, [7] skin, [8] cervical, [9] colon, [10] hepatic, [11] prostate, [12] breast, and renal [13]. The extract, seeds, and oil of *Nigella sativa* have proved to manage oxidative stress, hypertension, and diabetes, as well as [14] ulcers, [15] epilepsies, [16] fatty liver, [17] asthma, [18] arthritis, [19] inflammatory disorders, [20] cancers, [21], and parasitic diseases [22,23], in humans [24].

*Nigella sativa* is consumed in folk and Unani medicines in Pakistan. By following the previous literature, *Nigella sativa* has great potential for disease curing and health improvement; more research work has been needed to convert the herbal medicinal culture to new medicine systems. The thymoquinone contains a carbonyl polymer called Nigellon. Oil of *N. sativa* seeds and its active ingredients reveal therapeutic functions such as antiviral, antimicrobial, lowering the blood sugar level, antitumor, anti-oxidation, muscle relaxation, and anti-inflammatory [25,26,27,28]. Formerly, different kinds of chemical compounds were isolated from various species of *Nigella sativa* [29]. Hence, *Nigella sativa* has 84 g fiber, 216 g protein, 45 g ash, 38 g moisture, 406 g fat, 249 g free nitrogen extract, 60 mg zinc, 105 mg iron, 527 mg phosphorus, 15.4 mg thiamin, 18 mg copper, 57 mg niacin, 0.16 mg folic acid, and 1860 mg calcium per kg [30]. *Nigella sativa* is recognized as an annual herbaceous plant, which is included in the family Ranunculaceae and largely cultivated in different regions of southern Europe, as well as a few areas of Asia [31], which includes Saudi Arabia, Pakistan, Syria, India, and Turkey [32]. The colors of its flowers are mainly white, pink, yellow, light blue, or lavender, and its flower makeup has 6–10 petals. The fruity portion of the plant is a bulky and balloon-like capsule, which carries many black seeds with a bitter and aromatic taste [4]. The farming time for *Nigella sativa* falls between November and April, and its germination period is completed two weeks after seed sowing. However, the fruits are usually obtained from plants from January to April [33]. *Nigella sativa* seed oil and their active ingredients have been used in many dishes for chilling and flavoring [34]. About 28–36% fixed oil is present in *Nigella sativa* seeds, and it consists of a diverse range of unsaturated fatty acids, such as linolenic, arachidonic, linoleic, and eicosadienoic acids. In contrast, saturated fatty acids are myristic, stearic, and palmitic acids [35]. The other components of seed oil are citronellyl acetate, cholesterol, carvone, campesterol, α-spinasterol, stigmasterol, p-cymene, β-sitosterol, palmitoleic, oleic, citronellol, nigellone, and limonene [36]. The fixed oil contains 12.5% of oleic, linoleic, and palmitic acids; the volatile oils contain carvone, trans-anethole, limonene, and p-cymene [37]. The oil also contains considerable amounts of carbohydrates, amino acids, fixed or volatile oils, and proteins [32]. However, the versatility in the pharmacological properties of seeds is mainly due to the presence of quinine constituents, the most abundant of which is thymoquinone. The volatile oils of black seeds have larger quantities of thymoquinone. Gali-Muhtasib et al. [38] explained that thymoquinone, flavonoids, alkaloids, and tannins are the active ingredients of black seeds, extracted with ethyl alcohol and cold water [39]. Nowadays, *Nigella sativa* oil is categorized as functional oil because it has a high content of omega-9 (oleic, 15–24%) and -6 (linoleic, 54–70%) fatty acids, as well as others found in minor amounts [40]. This crude oil has a protective effect, mostly on nerve cells [41] and the liver [42]. In addition to other biological functions, *Nigella sativa* crude oil a carries small amount of volatile oil and exhibits functional properties, due to thymoquinone. This crude oil is safe and largely utilized in dietary supplements because it has less toxic effects [43]. The key constituent of oil is thymoquinone, which performs its function as an anti-epileptic agent [44]. This extract also contains tannins, terpenes, alkaloids, glycosides, saponins, steroids, and flavonoids [45]. Biologically active compounds of *Nigella sativa* are not stable during different chemical reactions, and their prescribed amount was not appropriate for clinical research. The *Nigella sativa* seeds also have unsaturated fatty acid esters with nigellimin, terpene alcohols, saponin, and the alkaloid nigellidine [46,47]. According to Agbaria et al. [48], the unroasted seeds have less anti-proliferative activity than the pretreated heated seeds (50–150 °C, about 10 min) for the milling process. Initially, the oxidation process was low; it was enhanced during storage (about 55 days) and leveled off [49]. To overcome all of the above-mentioned issues, different encapsulation techniques have been used for black seed oil and their active components. It is the most successful method for protecting thymoquinone (Figure 1). Today’s black seed oil is microencapsulated with emulsification processes spray-drying and nanoprecipitation. This encapsulated black seed oil has high phytochemical content, which improves the nutritional status of food items. *Nigella sativa* oil is consumed as a functional ingredient in food systems. It can be utilized in the form of flavoring and seasoning agents during food product development [50,51]. The objective of this research is to provide a narrative review of black seeds and its active ingredients, with different oil extraction methods and a characterization of different encapsulation processes, fatty acid profiles, oxidative stability, and medicinal properties. Black seed contain a high content of bioactive compounds, in order to define its intrinsic pharmaceutical and nutraceutical actions and provide future research directions for identifying novel drugs. Although other authors have previously reviewed some aspects of chemical properties, the effects of various pretreatments on its stability, and quality analysis studies, our review provides a more comprehensive analysis of the related studies.

## 2. Extraction of Oil from *Nigella sativa* Seeds by Using Different Novel Techniques

### 2.1. Cold Pressing

Oil can be extracted from *Nigella sativa* seeds by using the different methods. According to the Kiralan et al. [52], the cold pressing method is suitable for extracting *Nigella sativa* oil from seeds. In this method, mechanical pressing was used for the pressing of seeds at a temperature of 25 °C. Furthermore, the separation of oil and crushed seed fiber has been performed by soaking the solution for one night at a 25 °C temperature. After that, filtered oil was obtained by using a glass funnel and Watman #4 filter paper (0.45 μm, Vivascience AG, Hannover, Germany).

### 2.2. Supercritical Fluid Extraction

Another innovative method for the extraction of *Nigella sativa* oil from seeds was used by Mohammed et al. [53]. The supercritical fluid extraction equipment (FeyeCon Development B.V. Weesp, Netherlands) was used for *Nigella sativa* seed oil extraction, by using a stainless steel grinder (Waring Commercial, Torrington, CT, USA) for 3–4 min; the crushed dried seeds were obtained, placed the material in a 50-L container of extractor, and sealed tightly. The system used an automatic back pressure regulator for maintaining the temperature at 40 °C for 1 h; the pressure was 600 bar, and the flow rate of injected liquid carbon dioxide (CO_2_) was 150 L/h.

Rao et al. [54] also chose the supercritical fluid extraction method for *Nigella sativa* seed oil extraction. In its instrumentation, it contained a syringe pump with 260 mL capacity, controller system (ISCO 260D), and ISCO series 2000 SCF extraction system (SFX 220), consisting of a dual chamber extraction module with two 10 mL stainless steel vessels. Hence, about 5 g of ground black seeds were added in a stainless steel cell (10 mL). Then, the standard quantity of supercritical carbon dioxide (SC CO_2_) (50–400 mL) was flushed into the cell at a 1 mL/min flow rate. The final concentration of the extract was collected in the cold trap. After optimization of supercritical fluid extraction conditions, the lower yield of 0.84% (508 °C, 400 bar, and 100 mL) and higher yield of 31.7% (508 °C, 100 bar, and 200 mL) were obtained at optimum levels.

### 2.3. Soxhlet Extraction

Dinagaran et al. [55] used the soxhlet apparatus for *Nigella sativa* oil extraction from black seeds. For this purpose, *Nigella sativa* seeds were collected from different regions of India, including Tamil Nadu, Triplicane, and Chennai. During the sieving process, the small and contaminated seeds were removed at room temperature. In this process, the seeds were first ground using a tabletop mixture, hexane was used for extraction of seed oil for approximately 2 h in a soxhlet apparatus, and the extracted oil was stored at room temperature in a selected amber glass bottle until use. *Nigella sativa* seed has 28–35% fixed oil, which mainly consists of unsaturated fats. Through gas chromatography–mass spectrometry (GC-MS) analysis, 32 different compounds were found in black seeds.

### 2.4. Hydro Distillation (HD) Method

Kokoska et al. [56] selected the hydro distillation (HD) method for the extraction of oil from *Nigella sativa* seeds. In the first step, the seeds were ground at 25 °C. Then, they weighed the 70 g sample to be used for further analysis. The average yields were achieved and figured on a dry weight basis. For attaining essential oil through the HD method, they used a water holding flask for placing the material. It is called a Clevenger-type apparatus because the flask is directly connected to the condenser. After 2 h of continuous processing, a yield of 0.29 wt/wt of pale-yellow oil was obtained.

Burits and Bucar [57] also chose the same technique for oil isolation, and an Austrian pharmacopoeia (Clevenger apparatus) was used as standard apparatus in the whole process. The results were not satisfactory because the oil extracted had lower quantities of essential oil, with only 3% thymoquinone content, while Soxhlet extraction yielded 48% thymoquinone content.

### 2.5. Microwave-Assisted Extraction (MAE)

Abedi et al. [58] performed the oil extraction through a domestic microwave oven (Daewoo Electronics KOC-154KWR Microwave Oven) with a frequency of 2450 MHz. Initially, they took 50 g of ground seeds and selected a 500 mL round-bottomed flask for the soaking of seeds in 50 mL of water for about half an hour. After that, the Clevenger apparatus was fixed with a flask and utilized 450 W of power for heating (30 min). However, the essential oil was leached out in the n-hexane solvent. Only 0.33% essential oil yield was achieved by using MAE extraction conditions (power 450 W, moisture content 50%, and time 30 min).

### 2.6. Ultrasound-Assisted Extraction 

Moghimi et al. [59] used an ultrasound-assisted extraction method for oil extraction. For one treatment, a sample of 500 g was transferred to the 1.5-l container that was placed in the ultrasonic bath. Several optimization conditions were selected, including the time (30, 45, and 60 min) and ultrasound pretreatment power (30, 60, and 90 W) at a fixed frequency of 25 kHz. After completing this process, the oil was isolated by using a screw press at 33 rpm speed. The maximum results of 39.93% extraction efficiency were achieved at power of 90 W and time of 60 min, while the minimum results of 27.29% extraction efficiency were achieved at power of 30 W and time of 30 min.

### 2.7. Steam Distillation

For the prevention of the side effects of degradation, steam distillation was performed at a low temperature. In 100 mL of distilled water, 10 g of seeds were added and mixed. This mixture was quantitatively transferred into the separatory funnel. This process of extraction was performed three times; a total of 10 mL of diethyl ether was added at every step, and the funnel was shaken vigorously. Sodium sulfate was used to dry the organic layer, and 0.4% was the obtained yield after evaporation in the water bath [60]. The steam distillation process was used by Kokoska et al. [56]. A glass column-containing material was interpolated between the condenser and flask. The yield of oil that was extracted by steam distillation was 0.39%, and the color of the oil was pale yellow.

### 2.8. Accelerated Solvent Extraction (ASE)

A 1 g sample of black seeds in powdered form was taken in a stainless steel cell with a 34 mL capacity. The conditions were set: 100 atm pressure, 10 min static time, 20% rinse volume, 2 extraction cycles, 30 s purge time, and 26 mL of solvent volume. P1-P9 black seed samples from Pakistan, Indian, and Saudi Arabian were treated with n-hexane as P1-P3, methanol (MeOH), and dichloromethane (DCM) at 40 °C, P4-P6 with MeOH, DCM, and n-hexane at 50 °C; the same procedure was performed for P7-P9 at 70 °C. The results reveal that the solvent with high yield, following n-hexane, was MeOH, whereby the yield and recovery observed was 2.5 g (12.5%) for Saudi Arabia, 2.2 g (11%) for Pakistan, and 2.04 g (10.2%) for Indian black seed sample [61] (Table 1). 

## 3. Fatty Acid Profile of Extracted Oil

*Nigella sativa* seed oil, extracted from black seeds, contain a considerable amount of polyunsaturated fatty acids (PUFAs), with a minor amount of long-chain polyunsaturated fatty acids. The fatty acids composition of black seed oil contains the lauric (0.6%), myristic (0.16%), palmitic (11.4%), margaric (0.07%), stearic (3.2%), palmitoleic (1.15%), margaroleic (0.04%), oleic (8.1%), eicosenoic (0.4%), linoleic (55.6%), linolenic (2.45%), behenic (0.87%), erucic (1.0%), lignoceric (0.02%), eicosapentaenoic (5.98%), and docosahexaenoic (2.97%) acids. The results of another research work showed that black seed oil is higher in components such as linoleic (427.8 g/kg) and oleic (294.3 g/kg) acids, while it also contains other constituents, such as that of the lauric, myristic, myristoleic, pentadecanoic, palmitic, palmitoleic, heptadecanoic, heptadecenoic, stearic, elaidic, oleic, linoleic, linoleic, linolenic, arachidic, gadoleic, eicosadienoic, behanic, erucic, and lignoceric acids. Moreover, the fatty acid profile of black seed oil showed the concentration of linoleic acid to be the highest (58.9%), with prominent oleic (28.1%), palmitic (12.5%), and stearic (3.1%) acids, respectively [63,64,65]. In another recent research work, carried out by Farhan et al. [66], the fatty acid composition of black seed oil was made up of the myristic (0.24%), palmitic (11.10%), palmitoleic (0.23%), heptadecanoic (0.56%), stearic (2.60%), oleic (24.6%), linoleic (58.8%), arachidic (0.22%), linolenic (0.4%), eicosenoic (0.18%), eicosadienoic (0.22%), eicosatrienoic (0.74%), and behenic (0.11%) acids. However, black seed oil contains a high percentage of triacylglycerides, especially acyl groups of oleate and linoleate, which are considered essential and omega-rich sources. Black seed oil contains omega-3 (Ω-3), -6 (Ω-6), and -9 (Ω-9) fatty acids, as well as bioactive compounds, which are used to reduce the risk of cardiovascular disease, inflammation, hypertension, oxidative stress, hormonal disorders, chronic cancer, diabetes, neurodegenerative diseases, metabolic, and skin disorders, while also improving immunity and blood vessel circulation [67,68] (Table 2).

## 4. Oxidative Stability of Extracted Oil

The initial concentration of black seed oil primary oxidation products was very low, but enhanced, during the storage time of 55 days and then flattened. During this storage period of 55 days, nanoparticles of less encapsulated efficiency increased their peroxide values. The peroxide value of these nanoparticles was lower than the initial value of peroxide [49]. Due to the increase in temperature, the peroxide values also increased. The peroxide value of unencapsulated black seed oil was more than nanoparticles [75]. During storage, the secondary oxidation product formation was determined by the p-anisidine value [76]. In the unencapsulated black seed oil, the p-anisidine value was more than in nanoparticles, which showed that the effect of encapsulation is protective [77]. The totox values at the same temperature and storage time were observed. At 25 and 60 °C, the highest totox were determined in unencapsulated samples at the end of storage time. The increase in the totox value of the encapsulated samples was less because of the extra layer provided by the coaxial process [78]. The stability of black seed oil is improved using the various micro- and nano-encapsulation techniques at optimum conditions. It also provides many advantages, including improved thermal and chemical stability, preservation of the taste and flavor, controlled and targeted release, and the improved bioavailability of natural pigments [79].

### Technologies to Extend the Oxidative Stability of Black Seeds Oil

Different technologies have been used to improve the oxidative stability of black seed oil and its components, as presented in Table 3. The high concentration of thymoquinone is present in black seed oil; as reported by Ravindran et al. [80], this compound is sensitive to oxygen, light, heat, moisture, and food processing or storage conditions. The thymoquinone in black seed oil is protected by encapsulation. The microencapsulation is achieved by the emulsification process, [50] spray-drying [51], and nanoprecipitation [81]. These processes take more time and involve hot gas streams; therefore, they are not appropriate for bioactive substances that are heat sensitive [82]. For sensitive compounds, alternative techniques are required to overcome the drawbacks of the conventional method of encapsulation.

The recently developed process for encapsulation technology is electrospraying. It is an alternative to conventional processes. This process is cost-effective and simple to produce a variety of nanoparticles. There is no need for heat during drying, and the formation of smaller encapsulates of 1–5 m is gaining importance in temperature-sensitive bioactive encapsulation [83]. This process includes a polymer solution; its surface is charged by an electrostatic field of high voltage and produces ultrathin droplets at room temperature after evaporation of solvent, resulting in the formation of dried capsules of micro and nano sizes [84]. 

An alternative procedure can be used for the black seed oil encapsulation with zein. Zein is a prolamin protein that is water insoluble and obtained from maize. This protein is hydrophobic in nature and used in coating materials for food. It is reported as a favorable and protective matrix used in the electrospraying process for bioactive compound encapsulation [85]. The atomization of solution polymer is involved in this process by a pneumatic injector, and a high electric field is used for nebulization. In this procedure, the evaporation of the solvent, by air at room temperature in a drying chamber, results in free-flowing powder containing encapsulated material being collected [86]. The efficiency of the electrospraying process is increased by EAPG and used in the industrial production of omega-3 capsules that are electro sprayed and used for food application [87]. 

For the ease and protection of handling, it involves the incorporation of functional or sensitive core substances, which are called encapsulates in the wall material [51]. In this process, thin coatings are used on dispersions, small liquid droplets, and solid particles [88]. The particles range in size from 1 to 1000 μm and have a core that is coated with synthetic or natural shells called microcapsules [89,90]. These microcapsules are used for the protection of oils from environmental factors [91]. This microencapsulation increases the shelf life of black seeds. The microencapsulation also helps in the regulation of the release rate of the oil, in order to sustain the absorption and maintain an appropriate concentration for the production of the desired effects [92]. The encapsulation of oils can be achieved via various methods, but electrohydrodynamic atomization (EHDA) is one of the most important for the production of nano- and micro-sized structures [93,94]. Various factors, such as the nozzle size, applied voltage, flow rate, and distance of the collection of the polymeric solution that is required to produce the desired size of particles or morphology, must be taken into account [95]. In the microencapsulation process, an electrospray technique is also used for the fabrication of a wide range of particle sizes [96]. To overcome the viscosity and surface tension of the polymeric solution, which leads to the distortion of particles at the needle tip, electrostatic forces are applied [97]. A jet of high charge density is produced, and then an array of charged beads of sizes ranging from millimeters to micrometers is produced by the shattering of the jet [98]. These droplets drop into a solution for cross-linking. The electrospraying process has been applied for the preparation of particles that are synthesized from synthetic and natural polymers [99]. The economically feasible and most common process for the encapsulation of powdered ingredients is spray-drying [100,101]. Different investigations into microencapsulation by spray-drying have been performed to check its protective effects against oxidation [102,103,104].

The anti-oxidant and -bacterial activities of essential oils can be improved by the emulsification process [105]. An emulsion can be defined as the dispersion of an immiscible liquid into another liquid, and this dispersion is stable [106]. It is difficult to protect encapsulated material from leaking and changing in its composition when small molecules are allowed to penetrate through the wall material during a specific time period [107]. To overcome this problem, a pre-encapsulation process and solid wall material are recommended [108,109,110,111]. Emulsification can be achieved before encapsulation [112,113]. This emulsion-based system has developed from the encapsulation process of dispersion of lipophilic ingredients in aqueous media. Various processes have been used for emulsion preparation, such as high-speed mixers, colloid mills, ultrasonic homogenization, and high-pressure homogenizers [114]. In the encapsulation process, the efficiency and stability of average-sized particles is difficult to maintain because particles of larger size have better protective effects, as compared to smaller ones; however, in the food matrix, they present low dispersion [110].

Encapsulation is commonly used to achieve food ingredients or any other components with a diameter of 1–1000 microns. Furthermore, when such terms are fulfilled, this methodology can allow for the sustained release of the encapsulated core [115]. Fatty acids, as well as artificial ingredients and coloring agents, have all been encapsulated using a spray dryer [116]. Encapsulation is also extensively used in various food industries to integrate oil aromas in a spray-dried form because it is cost-effective, versatile, can be used in a continuous mode, and yields better particles [117]. The most widely accepted encapsulation technology, used throughout the food market to encapsulate omega-3 fatty acids (PUFAs), is spray-drying [118]. Despite the short holding time of encompass in the drying medium (up to 60–80 °C), the oxidation of omega 3 PUFA is caused by temperature increment of encapsulate (up to 60–80 °C) during the spray-drying application’s falling-rate time frame [119]. Freeze-drying, but at the other hand, which also works under room temperature (e.g., vacuum and low temperature), has a lower bandwidth, and is expensive. Furthermore, the magnitude of encapsulates produced by the spray- and freeze-drying mechanisms is quite huge (10–100 m), which may have an adverse impact on the final enhanced product’s organoleptic characteristics [120]. Spray-drying is accomplished by converting a slurry emulsification from a fluid to a powder in a running condition by solubilizing the core materials in water to produce an emulsification in fluid state and feeding this emulsion into the hot form of media (100–300 °C) to evaporate water. The dried item can be obtained as powder or flocculated particles [121]. Spray-drying encapsulation has four different steps: (i) creating a stable emulsion; (ii) homogenizing the dispersion; (iii) atomizing the emulsion; and (iv) dehydrating the atomized particles. To achieve maximum concentration of the polymeric chains and inhibit any variability induced by temperature variations, the very first step is usually carried out by solubilizing the wall materials in filtered water, as well as emulsification, or dissipating using a stirring rod 24 h at 25 °C [122]. Based on the emulsification properties of the wall materials, core materials are blended with an aqueous medium of the internal walls; then, the emulsification agent can be added before entering the second phase. The created emulsion, which contains the wall, as well as core components, should be stable until the drying phase [123]. Spray-drying is the most cost-effective technique of encapsulation, when compared to other encapsulation methodologies. The encapsulation of oils incorporates a range of variables, such as inlet and outlet temperatures, total dissolved solids, and the form of internal walls used, all of which have significant effects on the resulting product’s quality [124].

## 5. Functionality in Food Applications

Natural foods that are high in nutrients and may have biological activities are in high demand among modern consumers (Table 4). This needs to motivate food makers and studies to develop novel food formulations that are supplemented with various substances [76]. Several earlier researches suggested enhancing the nutritional value of food products by adding oils high in phytochemicals. *Nigella sativa* oil has been proposed as a useful component in food [129]. *Nigella sativa* (Klonji) is a valued seasoning, with a particular aroma and flavor, that has been used in pickles, baked goods, confectionary, sauces, salads, and savory foods [130]. The seeds have been discovered to be utilized as a seasoning and flavoring agent in Indian and Middle Eastern cooking [131]. *Nigella sativa* seeds are used as a spice in a variety of cuisines [132]. To make a bitter qizha paste, the seeds are crushed [133]. Curries, veggies, and pulses all benefit from the dry-roasted seeds. They can be used in recipes with pod fruit, vegetables, salads, and chicken as a spice. Black seeds are used as part of the spice mixture in several cultures [134]. Black seed is also used in tresses cheese, a Middle Eastern braided string cheese known as majdouleh or majdouli. Black seed (Klonji) is a plant that is used as a food preservative, as well as a medicinal powder [25]. The biological actions of black seed oil are attributed to the presence of phytochemical molecules, known as thymoquinone, according to Hassanien et al. [135].

*Nigella sativa* oil was utilized to fortify the ice-cream product, since oil-in-water nanoemulsion may readily be added to dairy products to boost their nutritional content. Because of their functional qualities in food processing, nano emulsions have been incorporated to a variety of food systems [136,137]. Nanoemulsion was used to incorporate *Nigella sativa* oil into an ice cream product. The NSO nanoemulsion improved the physical qualities of ice cream, as well as customer acceptance. Sensory evaluation of the various samples found that the ice cream sample with 5% nanoemulsion obtained the most acceptance from the panelists. The results showed that NSO nanoemulsion can be used to fortify ice cream [138]. As a result, it might be used as an innovative component in ice cream, with a wide range of black seed oil health benefits. However, because black seed oil has a bitter or strong peppery flavor in the mouth and may affect the taste of ice cream food products, it was recommended that the amount of black seed oil nano emulsion not be raised by 10% [71]. 

Mayonnaise is one of the world’s oldest and most extensively used sauces. Oil (70–80 percent), egg yolk, vinegar, and seasonings make up traditional mayonnaise. Mayonnaise is particularly sensitive to spoiling and auto-oxidation, due to its high oil content [139]. Consumers are increasingly interested in natural foods and preservatives, in order to live healthier lives. As a result, the food industry has been on the lookout for new and interesting spice flavors for ethnic and cross-cultural cuisines [140]. Because of their antioxidant characteristics [78] and health advantages, black seeds (*Nigella sativa*) and their crude or essential oils are widely employed in functional foods, nutraceuticals, and pharmaceutical products [135,141]. Mayonnaises supplemented with black seeds had superior oxidative stability, compared to mayonnaise made with SFO. The phenolic chemicals in black seeds, particularly thymoquinone, may play a role in oxidative stability. Using it in mayonnaise recipes as a natural antioxidant source could help reduce oxidation, hence improving shelf life and flavor [142]. 

Chicken meat is high in easily digestible proteins, including omega enrich, lipids with a high PUFAs content, vitamins, and minerals. The increased degree of PUFAs in chicken flesh increases oxidative processes, thus resulting in a quicker loss of meat quality, when compared to other varieties of meat. Physicochemical qualities, biochemical and physiological profiles, and food hygiene all suffer from lipid oxidation. Furthermore, undesirable bacteria, initiated from animals, slaughterhouses, processing, and high temperatures, can contaminate chicken meat. Herbs and spices have been used for curing meat to give it a distinct flavor and keep it from spoiling during storage [143]. Plants and their extracts have a good effect on meat quality, which has been attributed to their high antioxidant and antibacterial activity. There has not been much research into the effectiveness of black seed extract in the muscle food system. The antioxidant capabilities of black seed have been used to protect meat products from oxidation. Black seed extract has the potential to significantly extend the shelf life of raw ground chicken flesh. It also ensures that the microbiological quality of the chicken flesh is maintained. The extract of black seeds improved the oxidative stability and safety of meat, changed the color of the samples dramatically, and stabilized the pH. The antioxidant capacity of black seed extracts was high in the lipid and protein fractions, whereas the ABTS•+ radical scavenging activity of black seed extracts was much lower [144].

## 6. Health Benefits of *Nigella sativa*

*Nigella sativa* are considered seeds of blessing in Arabic folk medicines. *N. sativa* are widely used in treatment of communicable and non-communicable diseases. The common issues of health, such as the common cold, chronic migraine headaches, and mild abdominal pain can be cured with the *N. sativa* tea. Black cumin was also used by ancient Chinese for improved liver function. Medicinal plants, such as black cumin, are used for curing of respiratory track ailments, including asthma, bronchitis, and rheumatism. Across middle Asia and southern countries, the digestive system is the most important system of the body for overall health and immunity of the body. Black cumin can help with the overall improvement of digestive functions, with an astonishing potential to treat the malfunction of digestive system organs. This seed of blessing can improve anorexia among patients and enhance the digestibility of food. The extract of black cumin can treat diarrhea, indigestion, and vomiting. Moreover, *N. sativa* has the potential to treat menstrual pain, and it is helpful for amenorrhea management. Roasted black cumin is recommended for vomiting as a quick remedy. The anti-viral power of these seeds can enhance immunity and is used for seasonal viral ailments. The therapeutic intervention also includes anti-bacterial, -inflammatory, -septic, and -fungal conditions. The health benefits of black cumin oil for skin conditions includes curing eczema, alopecia, freckles, and leprosy [1,154,155].

The pharmacological and biological activities include a wide range of applications. The powerful anti-inflammatory extract of *N. sativa* can inhibit the pro-inflammatory cytokines, interleukin 1 beta, and interleukin 6 (IL-6). The consumption of 500 µL of *N. sativa* for a month can improve the condition of chest congestion and treat sneezing. This extract can ameliorate allergic arthritis and hypertrophy of sinusitis patients. The bio active compound thymoquinone (TQ) possess hepatoprotective and nephroprotective power. The thymoquinone in black cumin has a strong antioxidant potential and treats lipid peroxidation. This is the most important bioactive compounds found in black cumin for the protection of heart diseases. The antiproliferative and proapoptotic activities of black cumin is astonishing in the prevention of abnormal lipid levels. The cardioprotective effects of *N. sativa* protects myoblast cells; it also protects against toxicity [156,157].

The properties result in the cell arrest and apoptosis of mutant cells. The anti-carcinogenic activity modulates the mutant cell into self-suicide and prevents cancer cell metastasis. A different animal trial described the positive outcomes of black cumin seed oil in the protection from cardiotoxicity. Black cumin seed oil can treat cardiac tissue damage and lower oxidative stress. Moreover, human trials at the dosage of 500 mg explained the neuroprotective potential. The better concentration, with improved memory and enhanced cognitive abilities, are the marvelous positive outcomes of *N. sativa* Oil consumption [158]. In addition to all these therapeutic applications of black cumin, a rat study explained the mechanism regarding Parkinson’s treatment by reducing oxidative stress, along with an increased amount of superoxide dismutase in the midbrain. The immunity enhancing properties are not only confined to neuroprotection, but also ameliorate the joint inflammation and lower the level of C reactive proteins. Both humoral and cellular immunity can be modulated by the usage of the blessed seeds from the prevention to treatment of simple to complex conditions (Figure 2) [159].

### 6.1. Medicinal Properties of Nigella sativa

Besides its medical properties, *Nigella sativa* also possess numerous pharmacological and biological activities and have been used as nutraceutical purposes and pharmaceutical alternatives, similar to dietary supplements that claim to have physiological benefits that could protect against chronic diseases. The medicinal properties of *Nigella sativa* are shown in Figure 3 [160].

### 6.2. Antiviral Activity of Nigella sativa against Other Medically Important Viruses

Viruses require a living cell for multiplication and cause disease in those cells and individuals. Black cumin seeds can help in the treatment of viral infections. The notorious virus in the human population for adverse outcomes includes hepatitis C virus (HCV) and HIV [161,162].

#### 6.2.1. Antiviral Activity of *Nigella sativa* against Human Immunodeficiency Virus

A pilot study was conducted by Maideen and Mohamed [163] regarding the *N. sativa* seeds for patients of AIDS. AIDS is a life-threatening disease that starts from asymptomatic stage of disease and progresses to symptomatic after years. HIV is the main cause of this condition, which not only a treat to life in itself, but also includes opportunistic infections to the person. This virus directly attacks the immune cells and deteriorates the immunity of the patient. A concoction based on *N. sativa* and honey was given to 51 patients in that study for 5 months. The results showed that the concoction, along with anti-retroviral therapy, was found to be effective in the reduction of the viral load. It enhances the overall immunity of the patients and increases the immune cells. The concoction increased the differentiation of CD_4_ cells among patients. These results promote the blessing seeds importance for HIV treatment. Globally hepatitis c is a viral disease and affects more than 179 million people. Hepatitis, due to any virus, results in liver cancer and causes miserable life conditions [164]. A self-controlled pilot study conducted in Egypt explained the potential of hepatic cell protection. *N. sativa* seed oil capsules of 450 mg potency were given to 30 patients with hepatitis. The oil of blessing seeds significantly reduced the viral load from the baseline values. Additionally, the controlled blood sugar level, with increased overall antioxidant capacity, were seen among patients. Improvements were seen in the liver function tests, including aspartate aminotransferase (AST), alanine aminotransferase (ALT), lactose dehydrogenase (LDH), and serum alpha-fetoprotein (AFP). The overall health of patients was improved via a decrease in the viral load, replication, and oxidative stress. Hepatitis B is more dangerous and life threatening, according to WHO. Millions of people suffer from this virus and end up with hepatocellular carcinoma. *N. sativa* being anti-viral, hepatoprotective, immune-protective, and anti-inflammatory is the best solution, with multiple health benefits [165].

#### 6.2.2. Antiviral Activity of *Nigella sativa* against SARS-CoV-2

*N. sativa* is famous for its medicinal worth from centuries against pathogens. The application as anti-SARS-CoV-2 agents enlightened its worth during pandemic of COVID-19. *N. sativa* tea with honey can ease the symptom of COVID-19 and improve immunity [166,167,168].

### 6.3. Role of Nigella sativa against Asthma

Asthma is a chronic inflammatory disorder of the respiratory tract. The inflammatory cells involved in this condition are eosinophils, t cells, macrophages, neutrophils, epithelial, and mast cells. Different environmental factors that are involved in the worsening of asthma include seasonal allergies and dust. Inflammation may be present in both allergic and non-allergic asthma. Mast cells cause inflammation in the air ways among patients of asthma. Black cumin seeds inhibit inflammation. Thymoquinone can inhibit leukotriene formation in human blood, through the inhibition of various enzyme activity. Thymoquinone functions as an anti-oxidant, with anti-inflammatory and -allergic potency to treat airway hypersensitivity-related diseases, including asthma. Bronchodilation and anti-allergic actions of extract of blessing seeds was also proven in preclinical studies [18,169].

### 6.4. Antioxidant Activity and Bioactive Compounds of Nigella sativa

Inflammation and oxidative stress are the root causes of all kinds of diseases at certain level ranges, from flu to all chronic diseases. Reactive oxygen species, hydroxyl, and free iron, along with other free radicals, are normally produced in every cell of the body during normal metabolism. Toxins come with the normal diet eaten by a person. These toxins can contribute to an imbalance between the pro-oxidant and antioxidant level. This condition leads to various diseases. Antioxidants protect from the damage of free radicles and prevent metabolic and chronic conditions. They protect from inflammation and reduce the burden of disease by lowering the risk of pathogen attacks. Phenols and flavonoids are powerful antioxidants that prevent all kinds of free radicals, and *N. sativa* contain a variety of antioxidants [170,171]. In Virto research explains the different antioxidant activities of different extracts containing phenolic compounds. Another study showed syringic acid, p-coumaric acid, thymoquinone, and Vitamin E. The antioxidant effect of the black seed extracts contained astonishing usage in the traditional and alternative medicine [172]. Black cumin was used for the low testosterone level by ancient Arabic people. Black seed oil can improve muscle mass and stamina, as well as increase testosterone levels. These properties make this oil a treatment for infertility [173]. Aging can cause prostate enlargement and become a reason of urine incontinency among old age males. Black seed oil can treat prostatitis and provide relief from this condition [174].

## 7. Conclusions

There is a lot of evidence showing that black seed, black seed oil, and its active constituents have effective anti-oxidant, -microbial, -parasite, -viral, -fungal, -inflammation, and -diabetics properties against many disorders in humans and animals—it is a relatively safe food, with a long, remarkable history in the traditional medicinal system. It can be used as a drug in the pharmaceutical and nutraceutical industries, due to its anti-microbial properties. It is suggested that a new drug can be characterized and developed from the active ingredients of the black seed. For this purpose, researchers and scientists can apply many new technologies to reach that goal. Experimental or clinical research work is needed to further assess the effects on the body. Furthermore, industries should change their attitudes and strategies and invest in natural compounds that have great potential, according to their biological properties.

## Figures and Tables

**Figure 1 foods-11-02826-f001:**
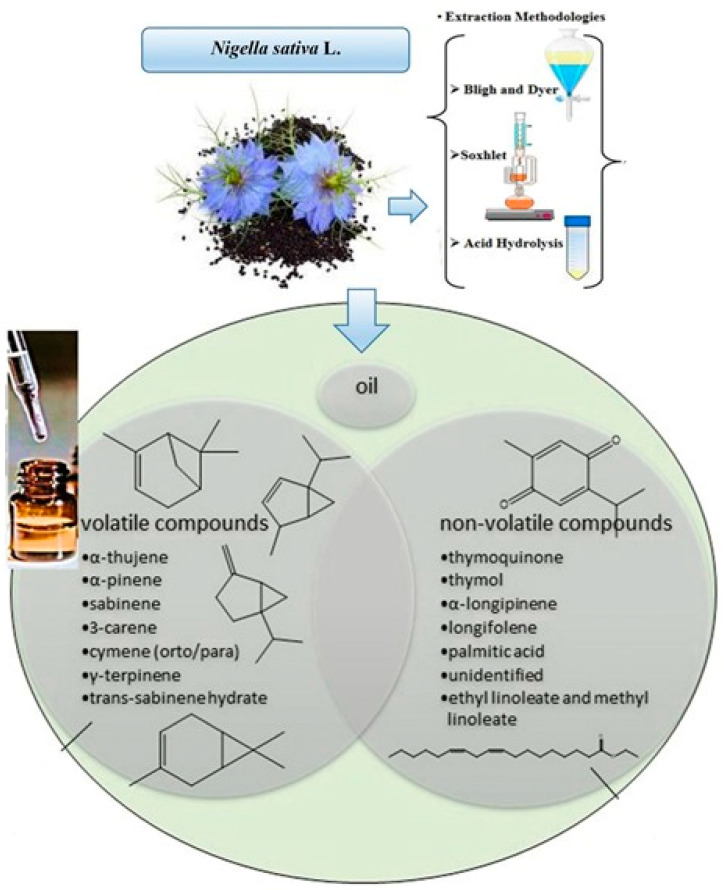
*Nigella sativa* oil extraction and its chemical composition.

**Figure 2 foods-11-02826-f002:**
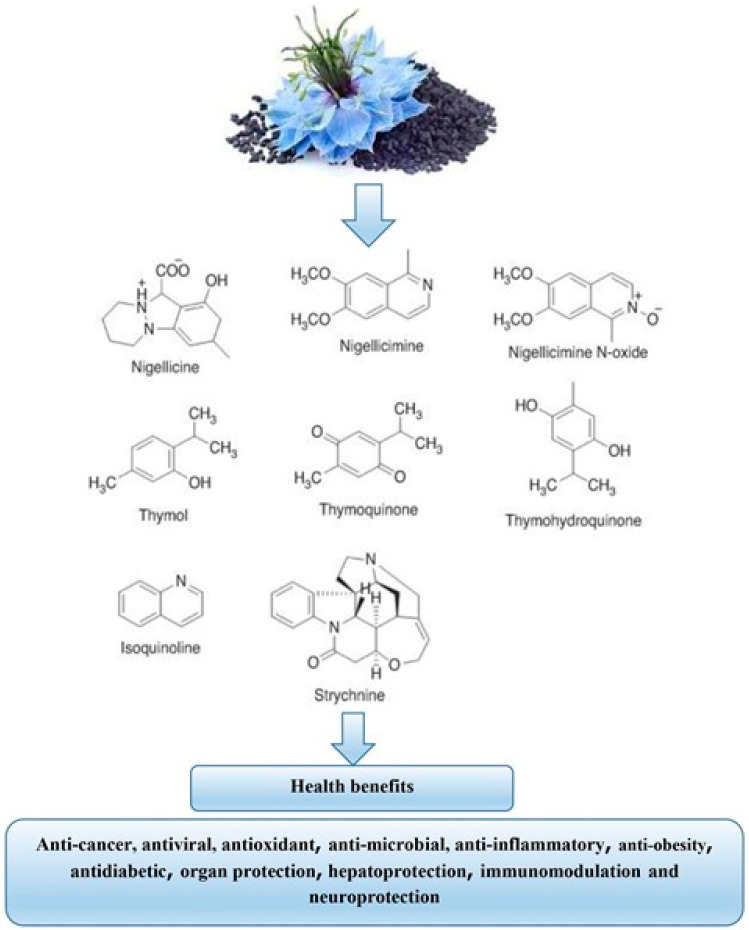
*Nigella sativa* health benefits.

**Figure 3 foods-11-02826-f003:**
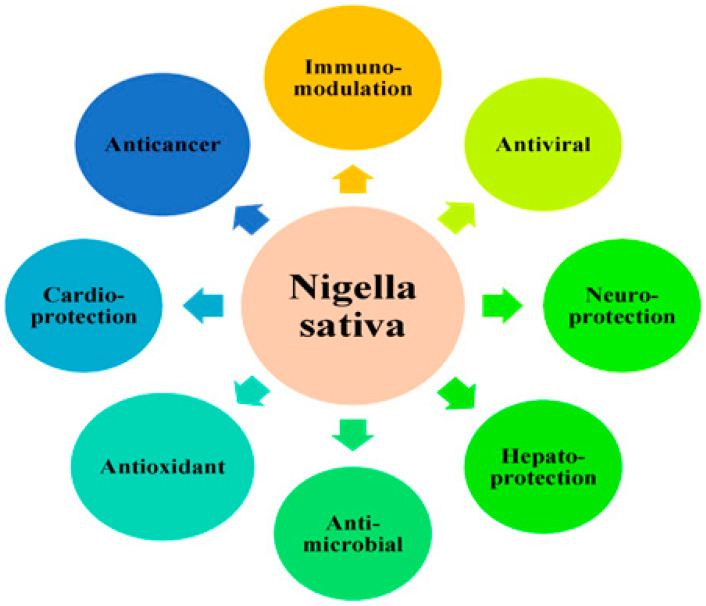
Medicinal properties of *Nigella sativa*.

**Table 1 foods-11-02826-t001:** Different extraction methods of *Nigella sativa* seeds oil.

Extraction Method	Solvent Used	Advantage	Disadvantage	Yield/Efficiency	Source
Cold pressing	Hexane	Involves no heat or chemical treatments during oil extraction	Provides low yield	27%	[62]
Supercritical fluid extraction	SC CO_2_	Rich in antioxidants	High cost	31.7%	[54]
Soxhlet extraction	Methanol	Low in cost	Residues of solvent has been left behind in the extracted oil	29.9%	[55]
Hydro distillation (HD) method	Water	Very simple method and instrument, shorter extraction time, free from organic components, less labor consumption, good in quality, lower cost with good efficiency	High energy is required for extraction	0.29%	[56]
Microwave-assisted extraction (MAE)	n-hexane	Free from organic solvent, less time with maximum yield	Additional filtration or centrifugation required to remove the solid residue	0.33%	[58]
Ultrasound-assisted extraction	Hexane	Less energy and solvent consumption, reduced time of extraction		39.93	[59]
Steam distillation	Sodium sulphate	Performed at a low temperature to prevent from degradation	More time consuming, due to the low pressure of rising steam	0.40%	[60]
Accelerated solvent extraction	MeOH, DCM, and n-hexane	A latest and efficient method for extraction		2.5 g, 2.2 g, and 2.04 g	[61]

**Table 2 foods-11-02826-t002:** Fatty acids profile of *Nigella sativa* seed oil.

Carbon Chain	Chemical Name	Procurement	Oil Extraction Method	Percentage (%)	Chemical Formula	Chemical Structure	References
Saturated fatty acids	
C_12:0_	Lauric acid	Iran	Hydrodistillation	0.6	C_12_H_24_O_2_	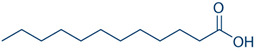	[37]
C_14:0_	Myristic acid	Middle East	Soxhlet extraction	0.16	CH_3_(CH_2_)_12_COOH	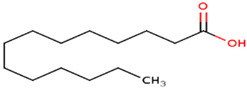	[69]
C_16:0_	Palmitic acid	Izmir	Steam distillation	11.4	C_16_H_32_O_2_	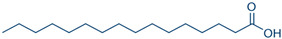	[70]
C_17:0_	Margaric acid	Konya, Turkey	Microwave-assisted extraction	0.07	C_17_H_34_O_2_	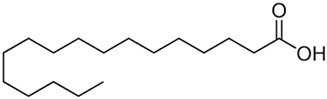	[52]
C_18:0_	Stearic acid	Moroco	Solvent extraction	3.2	C_18_H_36_O_2_	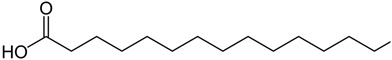	[62]
Monounsaturated fatty acids	
C_16:1_	Palmitoleic acid	Tunisia	Cold solvent method	1.15	C_16_H_30_O_2_	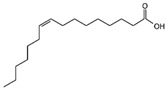	[71]
C_17:1_	Margaroleic acid	Turkey	Soxhlet extraction	0.04	C_18_H_30_O_2_	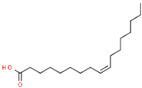	[52]
C_18:1_	Oleic acid	India	Solvent extraction	8.1	C_18_H_34_O_2_	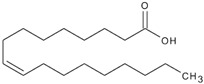	[72]
C_20:1_	Eicosenoic acid	Germany	Method ISO 659:1998	0.4	C_20_H_38_O_2_	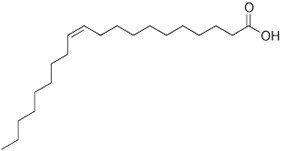	[73]
Polyunsaturated fatty acids (PUFAs)	
C_18:2_	Linoleic acid	India	Soxhlet extraction	55.6	C_18_H_32_O_2_	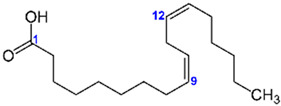	[55]
C_18:3_	Linolenic acid	Isfahan, Iran	Modified Bligh–Dyer method	2.45	C_18_H_30_O_2_	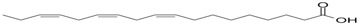	[74]
Long chain polyunsaturated fatty acids	
C_20:3_	Eicosatrienoic acid	Saudi Arabia	Soxhlet extraction method	0.74	C_20_H_34_O_2_	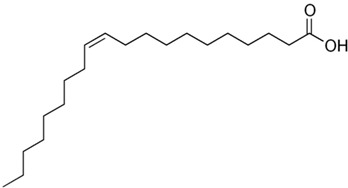	[66]
C_20:5_	Eicosapentaenoic acid	Iraq	Soxhlet extraction method	5.98	C_20_H_30_O_2_	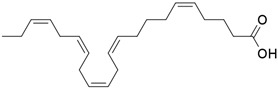	[64]
C_22:6_	Docosahexaenoic acid	2.97	C_22_H_32_O_2_	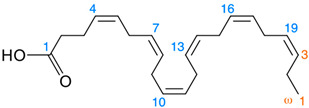

**Table 3 foods-11-02826-t003:** Technologies use to enhance the oxidative stability of black seed oil and their components.

Source/Component	Method	Capsulation	Efficiency/Yield	Peroxide Value	Size Detection	Particle Size	Conclusion	References
Thymoquinone	Nanoprecipitation	Nanoparticles	97.5%	-	Transmission electron Microscopy	150 and 200 nm	Thymoquinone nanoparticles showed that the higher bioavailability; therefore, it can be used as anti-proliferative, anti-inflammatory, and chemo sensitizing agents against many disorders in humans and animals.	[80]
Black seed oil	Ultra-sonication and spray-drying	Micro- and nano-encapsulation	-	100 μg Fe^3+^	Electron microscope	Microencapsulation = 250 nm to 400 nm; nanoencapsulation = 50 to 188 nm	The oxidative stability of encapsulated materials was improved during storage, and its stability was close to fresh oil.	[50]
Oleoresin	Spray-drying	Encapsulation	84.2 to 96.2%	-	Static light scattering instrument	3.08 to 11.84 μm	Microcapsules can be used as a functional ingredient in various processed food and meat products, as well as in nutraceutical and other pharmaceutical applications with maximum stability.	[51]
*Nigella sativa* seeds oil	Nanoprecipitation	Nanoparticles	70% to 84%.	-	Malvern particle size analyzer	230 to 260 nm	The stability of nanoparticles was significantly improved after 30 days of storage. These nanoparticles can be used to improve skin penetration and reduce systemic concentration.	[81]
Black seed oil	Electrohydrodynamic atomization	Encapsulation	67.201% to 104.50%	-	Olympus light microscope	Oil emulsion = 282.93 to 463.23 nm	The results revealed that the emulsion with lecithin exhibited the higher emulsion stability (3% and 1%).	[125]
Black seed oil	Freeze-drying	Encapsulation	63% to 87%	-	Inverse phase microscopy	173 to 382 nm	The stability of encapsulation showed that the liposomal preparation was stable at ambient conditions for one month.	[96]
Black seed oil	Electrospray technique	Microencapsulation	100%	-	Digital microscope	Less than 3 mm	This study indicated the palatability was significantly improved in encapsulation, without reducing thymoquinone stability.	[126]
Black seed oil	Coaxial electrospraying	Encapsulation	65.3% to 97.2%	19.50 to 30.57 meq O_2_/kg	Field emission scanning electron microscope	116 to 257 nm	Nanoencapsulation of black seed oil significantly improved the oxidative stability, due to form the coaxial structures.	[127]
Black cumin seed oil	Freeze-drying	Encapsulation	Plasmolyzed loaded yeast encapsule = 59.97%; non-plasmolyzed loaded yeast encapsule—39.18%		Scanning electron microscopy	-	The stability of encapsulated black cumin seed oil with yeast cell of S. cerevisiae was successfully increased in suitable condition, compared to black seed cumin oil and nonplasmolysed yeast cell.	[128]

**Table 4 foods-11-02826-t004:** Food applications of black seed and their oil.

Procurement	Type	Preparation Method	Product Name	Quantity and Ratios	Analysis	Results	References
Malaysia	*Nigella sativa* oil	Supercritical fluid extraction	Ice cream	3%, 5%, 10% 5%, 10%, and 15%	Physiochemical stability	*Nigella sativa* oil was significantly improved the stability of ice cream under optimize storage conditions.	[138]
Iran	Oil microcapsules	Goff and Hartel’s method	Ice cream	3 and 5%	Antioxidants and phenolic content	The results indicated that, in fortified ice cream, the resistance of melting, minerals, and activity of antioxidant significantly improved.	[145]
Mansoura	Black seed oil and powder	Dried hot-air oven	Ground mutton	Black seed oil (1%, 2%, and 3%): powder (2%, 4%, and 6%)	Antibacterial and antioxidant effect	The results obtained by using 3% black seed oil and 4% powder in meat showed a decrease in the bacterial contamination, due to their anti-microbial properties.	[146]
Poland	*Nigella sativa* oil	Convection–steam oven	Pork patties	1.88% and 3.76%	Sensory evaluation and antioxidant	*Nigella sativa* oil significantly improved the antioxidant activity of fortified patties.	[147]
India	Black seed oil	Ultrasound	Skim milk	7%	Physiochemical stability and size of emulsion	Stable emulsions of 7% black seed oil and milk were produced at lower time.	[148]
Polska	Black seeds	Freeze dried	Chicken meat	15 g	Antioxidant, phenolic contents, lipid oxidative stability, and microbial and sensory properties	The addition of prepared extract was significantly decreased the oxidation, while the stability of chicken meat was significantly improved during stored at 4 °C, due to good antimicrobial properties of black seed.	[144]
Turkey	Black cumin oil	-	Mayonnaise	5, 10, and 20%	Color, sensory, phenolic compounds, and oxidation	Mayonnaise formulated with black cumin oil was showed that the reduction in oxidation rate and improve the shelf life and flavor, as compared to other treatments.	[142]
Belgium	Black cumin seed oil	-	Bread	1 and 3 mL/100 g	Dough characteristics and bread analysis	Bread prepared with black cumin oil has been revealed to have good antifungal activity.	[149]
India	Black cumin seeds	-	Cookies	2%, 4%, 6%, and 8%	Chemical composition and sensory analysis	The results indicated that the good quality of product was prepared at 4% and rich in nutritional properties, as compared to other treatments.	[150]
Pakistan	Black cumin seed oil	Solvent extraction method	Cookies	1, 2, 3, 4, and 5%	Physicochemical, phenolic contents, peroxide value, and sensory analysis	Black cumin seed oil had a positive impact on the physicochemical, peroxide value, and sensory analysis of cookies, due to the good nutritional properties.	[151]
Egypt	Black cumin seed oil	Cold pressed	Soft cheese	0.1% and 0.2%	Microbial, physicochemical, sensory analysis	Black cumin seed oil was improved the nutritional profile of cheese and antimicrobial activity against pathogens.	[135]
Erzurum	Black cumin	-	Meat balls	0.50 and 1%	Physicochemical and antioxidant properties	The concentration of heterocyclic aromatic amines in fortified meatballs was significantly reduced, while the antioxidant properties and phenolic compounds was increased, as compared to control.	[152]
Turkey	Black cumin honey	-	Yogurt	0, 2.5%, 5%, 10%, and 15%	Phytochemical and antioxidants analysis	Results showed that, generally, the addition of black cumin honey in yogurt resulted in a significant increase of total phenolic contents and antioxidants activities.	[153]

## Data Availability

The data it is availability within this study.

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
