# Peer review of "A Narrative Review on Various Oil Extraction Methods, Encapsulation Processes, Fatty Acid Profiles, Oxidative Stability, and Medicinal Properties of Black Seed (Nigella sativa)"

_foods, 2022, doi:10.3390/foods11182826_

Round 1

Reviewer 1 Report

The MS is interesting and well organised, but some of the portions need modifications. The observations are as follows:

1. Line no 53-56 is not required.

2. L 61-64: no scientific significance

3. As mentioned by the authors there are several published reviews available on the Nigella Sativa L, therefore, the novelty of the current review needs to be addressed in the introduction.

4. The extraction methodologies should be explained under sub-heading, their advantages and disadvantages should also be covered.

5. CO2: check

6. Line 223: report the quantity in percentage.

7. A section on encapsulation of ( nano and microencapsulation techniques) Black seed extract need to be included, the following article may help to get the details of the techniques https://doi.org/10.1007/s12010-021-03631-8

8. More recent and relevant references need to be added.

9. English editing is required.

Author Response

Comments and Suggestions for Authors

Dear Authors,

The review article entitle " A Narrative Review on Various Oil Extraction Methods, Encapsulation Processes, Fatty Acid Profiles, Oxidative Stability, and Medicinal Properties of Black Seed (Nigella Sativa L.) " aims to study the effects of extraction methods OF black seed (Nigella sativa L.) on oxidative stable in different food products. This study is well written and compiled. The content flow is smooth and easy to read.

Response: We appreciate the reviewer for the valuable comments.

The manuscript can be the following minor revision.

Introduction:

Line 83: Cite a few of the research to explain the importance of chemical modifications for authenticating a scientific sentence.

Response: Citations have been added according to the suggestion of the reviewer.

I suggest adding a reference for black seed production.

Response: Reference has been added according to the suggestion of the reviewer.

Kindly re-write the last paragraph of the introduction to be clear.

Response: Action taken according to the suggestion of the reviewer.

The conclusion

The conclusion is very concise.

Response: We appreciate the reviewer for the valuable comments.

Comments and Suggestions for Authors

The MS is interesting and well organised, but some of the portions need modifications. The observations are as follows:

  1. Line no 53-56 is not required.

Response: Action taken according to the suggestion of the reviewer.

  1. L 61-64: no scientific significance

Response: Changes have been made according to the suggestion of the reviewer.

  1. As mentioned by the authors there are several published reviews available on the Nigella Sativa L, therefore, the novelty of the current review needs to be addressed in the introduction.

Response: Novelty has been added according to the suggestion of the reviewer.

  1. The extraction methodologies should be explained under sub-heading, their advantages and disadvantages should also be covered.

Response: Sub-heading have been made according to the suggestion of the reviewer.

  1. CO2: check

Response: Checked

  1. Line 223: report the quantity in percentage.

Response: Action taken according to the suggestion of the reviewer.

  1. A section on encapsulation of ( nano and microencapsulation techniques) Black seed extract need to be included, the following article may help to get the details of the techniques https://doi.org/10.1007/s12010-021-03631-8

Response: Action taken according to the suggestion of the reviewer.

  1. More recent and relevant references need to be added.

Response: More recent and relevant references have been added according to the suggestion of the reviewer.

  1. English editing is required.

Response: The article has been edited by the senior colleague.

Reviewer 2 Report

Dear Authors, 

The review article entitle " A Narrative Review on Various Oil Extraction Methods, Encapsulation Processes, Fatty Acid Profiles, Oxidative Stability, and Medicinal Properties of Black Seed (Nigella Sativa L.) " aims to study the effects of extraction methods OF black seed (Nigella sativa L.) on oxidative stable in different food products. This study is well written and compiled. The content flow is smooth and easy to read.

The manuscript can be the following minor revision.

Introduction:

Line 83: Cite a few of the research to explain the importance of chemical modifications for authenticating a scientific sentence.

I suggest adding a reference for black seed production.

Kindly re-write the last paragraph of the introduction to be clear.

The conclusion

The conclusion is very concise.

Sincerely

Author Response

(The authors gave the same response as above.)
